

**GC Insights: The *Anthro-Pokécene* - Environmental impacts**
**echoed in the Pokémon world**
Lewis J. Alcott[1,2] and Taylor Maavara[3]
[1]School of Earth Sciences, University of Bristol, Wills Memorial Building, Queen's Road, Bristol, BS8 1RJ,
United Kingdom
[2]Ecohydrology Research Group, Department of Earth and Environmental Sciences, University of Waterloo,
Waterloo, Ontario, N2L 3G1, Canada
[3]School of Geography, University of Leeds, Leeds, LS2 9NH, United Kingdom
Correspondence: Lewis J. Alcott (lewis.alcott@bristol.ac.uk)

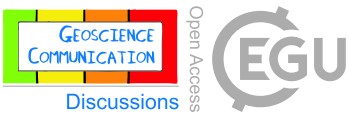

**Abstract**. Public perception of anthropogenic environmental impacts including climate change is primarily driven
by exposure to different forms of media. Here, we show how the Pokémon franchise, the largest multimedia
franchise worldwide, mirrors public discourse in the video games' narratives with regard to human impacts on
environmental change, demonstrating a trajectory towards greater and more explicit acknowledgement of climate
change and anthropogenic impacts in each released game.





**Introduction**

The perception and societal importance of anthropogenic impacts, including climate change, has evolved over
recent decades. This overall perception is both shaped and reflected not only by political discourse and news
media, but also by creative and narrative media, with ubiquitous blockbuster movies, television series and popular
literature illustrating climate and environmental change (Bulfin, 2017; McCormack et al., 2021). Video games
take over 3 billion players to virtual worlds where they can assimilate information as they see and interact with
virtual environments (Bankhurst, 2020), and have been recognized for their potential to teach and expose players
to concepts for decades (Adams, 1998; De Freitas, 2018; Squire et al., 2008). An investigation into Earth and
environmental science's representation in video games is still a growing field (Clements et al., 2022; Hut et al.,
2019; McGowan & Alcott, 2022; McGowan & Scarlett, 2021), with many video games taking place in
environments that are based on real world settings, events or locations, making them ideal settings to facilitate
learning related to environmental features, processes and interactions. In many cases, the graphical quality of
games has made it possible for game environments to be indistinguishable from their real-world counterparts (Hut
et al., 2019).

Pokémon is the largest media franchise worldwide with a total revenue near $100 billion USD (Bulchoz, 2021),
with 122 games including 36 main series games, merchandise, trading cards, numerous theatrical film releases
and a TV series spanning decades (ThePokémonCompany, 2022). Through gameplay, players can explore
interactions between anthropogenic and natural settings, showcasing and exposing human impacts on ecosystems,
both local and global, to audiences of all ages. As is well documented, climate change is a global challenge, and
with Pokémon media available across 192 countries (ThePokémonCompany, 2022), it is uniquely poised to be a
valuable resource as a climate change knowledge distributor. In doing so, we ask the questions: how have the
Pokémon video game's representations of environmental change evolved over the past three decades, and how
have they mirror public discourse and priorities?

**Methods**

We played and/or read game scripts of all the main series Pokémon games released from 1996 to 2023, to interpret
the overall narratives and design and compare how they have evolved through time. We additionally queried the
online Pokémon database Bulbapedia (Bulbapedia, 2023) with the following search terms for individual Pokémon:
endangered, climate, extinct, environment, ecology, ecosystem, adapt, hunt, extinct, fishing, and pollution/pollute.
We then compared them against the timeline of public perception and growing acknowledgement of
anthropogenic change and major events in climate policy, benchmarked using IPCC Assessment Reports and
major UN decisions including the signing of the Kyoto Protocol and the Paris Agreement.




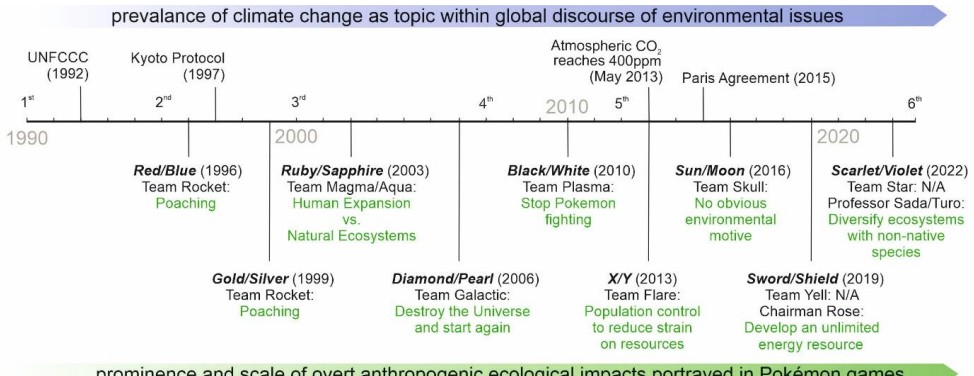

**Figure 1: Timeline showing the original release dates of the main-series Pokémon games (the earlier Japanese release dates are given for the first three games). As an example of the escalation of anthropogenic impacts portrayed in Pokémon games, summaries of the antagonists' motives are provided in green and how they relate to a human impact context. Above the timeline there are key events that have occurred since 1990 including the numbered IPCC Assessment Reports and key UN climate change agreements, which we show to benchmark the general trajectory of climate change as a topic and growing priority within global discourses and decision-making.**

**The Anthro-Pokécene through time**

The modern geologic era is often referred to as the Anthropocene due to widespread human impacts across geologies and ecosystems, caused by human impacts including climate change (Waters, 2016). The extent that the Anthropocene is represented in the Pokémon main series games reflects prominent topics within real-world public discourse. We thus refer to the era of anthropogenic change portrayed in the Pokémon world as the Anthro-Pokécene. The first four main series generations (*Red/Blue/Yellow*, *Gold/Silver/Crystal*, *Ruby/Sapphire*, and *Diamond/Pearl/Platinum*), released between 1996 and 2006, represent some elements of anthropogenic change, but these are largely limited to minor game script comments, Pokédex entries, or weak inferences that players could draw from game details, like the villainous "nefarious team" plotline (e.g. Team Rocket's efforts to poach Pokémon). These games coincided with a time in history when climate change was not the most central environmental topic in virtually all discourse that it is today (Observatory, 2023). In the 1990s, anthropogenic impacts to ecological systems that were often highlighted included poaching, overhunting, overfishing, and habitat destruction via deforestation and industrial pollution, which were in turn the issues highlighted in these early games. All the game development for *Red/Blue/Yellow* was completed before the Kyoto Protocol was signed in 1997, which represented a major step in terms of bringing climate change into the public awareness (Fig. 1).

The "nefarious team" plotline of the first game following the Kyoto Protocol, *Ruby/Sapphire* (2002), represents a real-world conflict based on the Isahaya Tidal Flats in the Japanese region Kyushu, which began in 1997 when the flats were drained to increase arable land area for agriculture (Kaliroff, 2022). The game represents the parties involved in this dispute as two antagonistic teams wishing to expand agricultural land or support marine biodiversity and health by expanding aquatic areas. This storyline was one of the first instances where the Pokémon franchise presented a morally ambiguous dilemma related to environmental change, whereby both parties were inherently trying to do the "right thing". The short period of time between when the conflict occurred



and the game's production highlights how the developers were paying attention to present day events and choosing
to represent them in the game.
More recent games acknowledge real-world environmental issues more directly, especially in games set in Alola
(*Sun/Moon/UltraSun/UltraMoon; 2016*) and Galar (*Sword/Shield, 2019*), which depict contrasting environmental
situations in ways accessible to a general audience. These generations of games were released following the
signing of the Paris Agreement in 2015 (Fig. 1), a time when the global environmental discourse had become
vocally aware of the urgent need to address the climate emergency. The former region, Alola, is a Hawaiian island-
inspired environmental utopia with a rich ecological diversity due to endemic island species. The latter, Galar, is
an UK inspired industrialized region in which the implications of pollution are evident. The most overt
representations of anthropogenic influence in the franchise arose in Galar. The coral Pokémon Corsola, previously
depicted as a healthy pink coral, appears in Galar as a white bleached coral, as the "living" version was wiped out
by ocean acidification driven by climate change.
The franchise's use of morally ambiguous storylines to present the nuance and complexity of environmental
change and associated decision-making in an increasingly politically polarized world. This trend is also found in
the earlier 6[th] generation games (*X/Y, 2013*), with a more extreme example of ambiguity: the antagonist wishes to
return the planet to a beautiful and unspoiled state. While arguably well-intentioned, the plan includes wiping out
most of the world's population to lessen the pressure on the natural world. This storyline mirrors the fraught real-
world argument that overpopulation is a root cause of climate change. Without being sanctimonious or forcing a
message upon players, the enemy inherently causes players to question the ethics of calls to reduce human
populations as a viable solution to climate change. The conclusion of this story notes that in order to create a better
world, people must cooperate globally, which is often quoted as a necessary approach to lessen climate impacts,
with the COP26 meeting being subtitled *Together for our planet* (TheUnitedNations, 2021).

**A hopeful world**
While the Pokémon franchise excels in its presentation of complex environmental situations to a varied audience,
the games notably present an overall hopeful representation of society's ability to respond to environmental
change. The games have transitioned from including polluting power plants (*Red/Blue, 1996*) to renewable energy
solutions such as wind farms (*Diamond/Pearl, 2006*), solar power (*X/Y, 2013*) and geothermal energy production
(*Sun/Moon, 2016*). This transition is not restricted to the progression of generations of Pokémon games; the
remakes of *Gold/Silver* (1998) named *HeartGold/SoulSilver* (2010), saw the introduction of wind turbines across
the region, ultimately leading to their widespread depiction in the most recent game *Scarlet/Violet*. The games
also include cycle paths and wildlife protection zones to demonstrate how the player can respect the environment.
Without ever needing to think critically about the game plotlines, in playing the games and remakes released since
~2010, players are moving through and interacting with worlds that represent examples of sustainable, often fossil-
free, living.
For many, Pokémon is a gateway to appreciating the natural world and understanding the scope and complexity
of responding to environmental change. Whilst we have noted examples of negative human-ecosystem



interactions, the Pokémon games expose players of all ages and demographics to ecological and environmental
concepts, likely many for the first time. Notably, Pokémon presents a hopeful balance between humans and the
environment, which is a rare depiction in an age of nihilistic, post-apocalyptic games and stories. Maintaining
hope that we can overcome modern environmental challenges if we want to continue to push for improvement,
rather than collectively default to hopeless catastrophism. Games and global phenomena such as *The Last of Us*
and *Fallout* are incredible and ground-breaking, but we need its antithesis in the world too, and Pokémon
represents that. Chang (2019) aptly summarizes this sentiment:

"*Given the present, fraught historical moment, in which scientists, activists, and educators are often*
*stymied in their efforts to depict the scope and urgency of global environmental crisis, games remain*
*largely untapped in terms of their potential to create meaningful interaction within artificially intelligent*
*environments, to model ecological dynamics based on interdependence and limitation, and to allow*
*players to explore manifold ecological futures— not all of them dystopian.*"






**Data Availability**

All data were collected through bulbapedia.bulbagarden.net and the game scripts as described in the Methods. Additional background information about the game can be found at https://corporate.pokemon.co.jp/en/ (last access: 6 December 2022, The Pokémon Company International, 2023). We do not have permission from the developers to share free access to the game. However, it is publicly accessible to purchase.

The authors explicitly state that they have no commercial ties to The Pokémon Company, Nintendo corporation, and/or its affiliates. This manuscript depicts work from a copyrighted video game or otherwise copyrighted material. The copyright for it is most likely owned by either The Pokémon Company, Nintendo and/or its affiliates or the person or organization that developed the concept.

**Author Contribution**

Both authors contributed to all aspects of the manuscript.

**Competing Interests**

At least one of the (co-)authors is a member of the editorial board of Geoscience Communication

**Ethical Statement**

The work presented is original and reflects the authors' views. Ethics approval and informed consent were not sought; this study does not deal with sensitive data or human participants.

**Acknowledgement**

TM was supported by an Independent Research Fellowship from the United Kingdom's Natural Environment Research Council (NERC), grant number NE/V014277/1.



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

*Science*, *351*.
