# Peer review of "GC Insights: The Anthro-Pokécene - Environmental impacts 1 echoed in the Pokémon world 2"

_Geoscience Communication, 2024_

## Author Response (AR1)

**Response to Reviewer 1:**

This is an insightful paper that leaves me wishing for a more extensive investigation on this topic. Assuming that in this case there are space limitations, I encourage the authors to expand and deepen their analysis in future publications. Similarly, given more space, I would recommend a deeper engagement with the games and nature/environment/ecology/climate change literature, which has been growing steadily in the past five years or so.

We would like to thank the reviewer for their overwhelming positive reception to the paper as well as the direction we hope our work will continue to go. As the reviewer points out, due to space limitations we were restricted with regards to the amount of content to include and would like to acknowledge that ongoing and future work will expand upon this piece.

Find below some notes that may improve the paper:

The authors say "In many cases, the graphical quality of games has made it possible for game environments to be indistinguishable from their real-world counterparts (Hut et al., 2019)." I do not think that this sentence is accurate, especially given that the reference provided does not seem to indicate anything of the sort, as it concludes: "We have demonstrated that while geoscientists might be slightly, but statistically significantly, better at separating real world photos of landscapes from video game screenshots, non-geoscientists are still capable of identifying landscapes from a video game, even when both the real world photos and the video game screenshots are filtered through an artistic "van Gogh" filter. This suggests that people recognize the natural features in video game worlds for the fantastical settings that they are..." I would suggest that the sentence is either clarified and properly justified with a reference (what is meant by indistinguishable? In terms of photo-realism? Plausibility of visible geological features? Complexity of ecological modeling?) or dropped altogether.

After feedback from the editor and the reviewers we have heavily revised the text content and made the overall message clearer. In this effort to streamline the paper, this section has now been removed.

"... how have they mirror public discourse and priorities?" -> mirrored

This has been corrected.

In the methods, please clarify what is meant by "game scripts." Are these complete transcripts of all the textual content that players encounter, summaries...?

As we have edited the content of the manuscript, we now only consider the narratives of the games and no longer include results of the game script analyses. We have edited the statement so that it now simply reads "To answer this, we played the main series Pokémon games released from 1996 to 2023, and thematically analysed driving narratives as well as notable instances of anthropogenic impacts in the games (Bulbapedia, 2024)."

"The coral Pokémon Corsola, previously depicted as a healthy pink coral, appears in Galar as a white bleached coral" -> I would also add that its type changes from water and rock to ghost, quite overtly implying that the coral is not only bleached but dead.

After this sentence we have now added: "*For example, the coral Pokémon Corsola, previously depicted as a healthy pink coral, appears in Galar as a white bleached coral, and changes from rock and water type to ghost type, as the "living" version was wiped out by ocean acidification driven by climate change.*"

"The franchise's use of morally ambiguous storylines to present the nuance and complexity of environmental 100 change and associated decision-making in an increasingly politically polarized world." -> Revise grammar (perhaps "the franchise uses").

The sentence has now been edited to read: "*The franchise goes on to use ever-growing morally ambiguous storylines to present the nuance and complexity of environmental change and associated decision-making in an increasingly politically polarized world.*"

"The conclusion of this story notes that in order to create a better world, people must cooperate globally, which is often quoted as a necessary approach to lessen climate impacts, with the COP26 meeting being subtitled Together for our planet (TheUnitedNations, 2021)." -> Beyond the framing of conferences and their messaging, the authors could cite the latest IPCC synthesis report and its insights on climate justice and cooperation as the mitigation and adaptation strategies advocated for.

We have further expanded this sentence to include a comment on the IPCC synthesis reports.

"*… with the COP26 being subtitled Together for out planet (TheUnitedNations, 2021), and cooperation being explicitly cited as a means of climate resilient development in recent IPCC reports (IPCC, 2023).*"

"Maintaining hope that we can overcome modern environmental challenges if we want to continue to push for improvement, rather than collectively default to hopeless catastrophism." -> Revise grammar, since I am not sure this expression works as a standalone sentence.

The sentence has been revised as follows: "*The existence of these utopian games promotes and maintains hope that we can overcome modern environmental challenges if we want to continue to push for improvement, rather than collectively default to hopeless catastrophism.*"

**Response to Reviewer 2:**

This paper poses an interesting question of increased environmentalism in the Pokémon franchise but fails to move beyond conjecture in answering it. I strongly encourage the authors to edit their methodology and reveal the results of their currently described methods (e.g., results of searching environment-related terms in Bulbapedia) in order to provide evidence of escalation of environmental thoughts in Pokémon and overall video game industry. This paper would also benefit from a wider focus beyond only main English-language video game releases to include shows, movies, and other pop culture impacts to definitively analyze increasing environmentalism within the Pokémon brand.

We appreciate the reviewer's interest in our paper. We do acknowledge that due to text limitations we aren't able to further expand beyond this main series of Pokémon games. Further investigation into non-English-language releases as well as the video game industry as a whole seems like a very worthwhile avenue for future work.

With regards to revealing the results, based on feedback from the reviews and editors suggestions, we have heavily edited the text of the manuscript and no longer collect data in this way, and instead purely discuss narratives in the games.

Notes that would improve the paper (specific edits, grammar,  etc…)

Abstract, line 17: Is your focus on Pokémon the franchise, or "video game narratives"? Both are mentioned throughout the paper, but only video games appear in Figure 1.

We mainly focus on the main-line of videogames. We would also like to note that the other Pokémon distributions (TV, film, cards etc.) are all based on the preceding video games so their initial development is used as the point of time comparison.

Abstract, line 18: You state you find both "greater and more explicit acknowledgment" – no results are shown to support this. Results that would be convincing are 1) more occurrences of environmental-related terminology in video game scripts or articles, and/or 2) more frequent (percentage of words) environmental references.

The terminology used here has been amended to simply state "*greater acknowledgement*". As mentioned we have now removed the approach previously taken with regards to keywords and now simply address the narratives in the games.

Introduction, line 43: Should likely be "have they mirrored public discourse".

This has been corrected.

Methods: what defines a "main series" Pokémon game? How were the search terms chosen? How were queries performed (e.g., exact match, entry heading only, separate words vs combinations, etc…)? How are these search terms compared across timelines of public perception and climate policy? How is public perception of anthropogenic change measured? Why are major events in climate policy limited to IPCC Assessment Reports and UN decisions? There are many country-specific rulings that could be used here, including Japanese policy changes that may affect Pokémon development more than UN-specific decisions.

The main series games are defined as the core series of games within the franchise. We have clarified this and provided a reference (https://bulbapedia.bulbagarden.net/wiki/Core_series).

As mentioned above, due to heavy revisions after reviewer and editor comment we no longer include a study of the search terms.

Public perception of change isn't directly measured. We assume that discussions around UN negotiation periods and IPCC reports drive a broad perception increase. Because of this we limit our findings to these two primary events as they are generally thought to be the two largest entities focusing on this topic.

While we acknowledge that as the games are made in Japan, Japanese policy may impact the developers perception of the games. However, as far as our understanding goes, the Act on Promotion of Global Warming Countermeasures that Japan follow is a result following the Kyoto and Paris Protocol guidelines (https://www.iea.org/policies/277-act-on-promotion-of-global-warming-countermeasures; https://www.amt-law.com/asset/pdf/bulletins12_pdf/210623.pdf). The act is due to the reviewed and updated, again in line with the Paris Agreement (https://news.un.org/en/story/2020/10/1076132).

Methods, line 51: Should likely be "anthropogenic climate change"

The statement has been removed during text edits.

Figure 1: Unclear on the relation between the antagonist motivations and climate-change related policies and events. Also, what scale are you using to define "escalation"? It is unclear how the 2006 motivation of "Destory the Universe and start again" has a lower escalation measure than the 2010 motivation of "Stop Pokémon fighting", for example.

We are not necessarily trying to demonstrate that there is a lesser escalation, but we are trying to demonstrate that conversations and narratives around climate change are becoming more prevalent in the games. We have amended this terminology and instead of escalation, have referred to the acknowledgement of anthropogenic impacts.

Figure 1: "Pokemon" in Black/White(2010) label should be Pokémon

This has been corrected.

Anthro-Pokécene, line 72: Observatory reference does not include pre-2004 papers, which therefore invalidates the claim "climate change was not the most central environmental topic in virtually all discourse", as there is no data in the reference to support or deny this.

We now provide the following reference to discuss pre-2004 environmental concerns.
https://www.enotes.com/topics/social-political-change-modern-america/questions/what-were-some-environmental-issues-during-1990s-343179

Anthro-Pokécene, lines 75-76: The Red/Yellow/Blue series is singled out for being different because it was completed before the Kyoto Protocol, but in Figure 1, it has the same theme as the Gold/Silver series. Why do you distinguish between pre- and post-Kyoto game development in this case?

This is true that it is difficult to gauge the correlation between the release date and the Kyoto protocol. However, we have now noted that the initial development of Gold/Silver began prior to the Kyoto protocol. Gold/Silver were originally announced in 1997 but was delayed a further 2 years due to development issues. Therefore the development of the narrative of the game likely took place prior to the Kyoto protocol.

Anthro-Pokécene, lines 78-86: The Kaliroff citiation, in addition to being an un-peer reviewed opinion article, does not include any direct quotes or other direct evidence linking Isahaya Tidal Flats to Ruby/Spphire game development. I agree there could be a link between the two, but the direct relationship implied by this paragraph is misleading.

In order to streamline the paper following the editors guidance, this section is now no longer present in the paper in its current form.

Anthro-Pokécene, lines 88-97: Once again, I agree that there is an environmentalist leaning to the Alola and Galar settings, but the link between them and the Paris Agreement is circumstantial at best.

We appreciate the concern here but we feel the reviewer may be misunderstanding the main message of our paper. We are not saying there is a direct link between the Paris Agreement and the games, nor are we even sure how it would be possible to make this claim. E.g. we make no claims in the paper that states that the Paris Agreement came out and the developers immediately responded by writing X lines in the script. This is unquantifiable. The link between the plot of the Pokémon games following the release of Paris Agreement and what is included in the agreement is circumstantial, and we make no claims otherwise.

The argument we are making is that globally, the impacts of climate change are increasing and in response, society as a whole discusses these issues more, and this discussion becomes more prominent over time in the Pokémon video games. The acceleration of climate change is well documented; it has been quantified in tens of thousands of scientific articles in fields ranging from atmospheric climate modelling to forest ecology to aquatic biogeochemistry to socioeconomics. As a result of the growing impacts of climate change, the global discourse about climate change has evolved and accelerated as well, because when something impacts people's lives, they talk about it more, and climate change is impacting everyone's lives. It is not possible to reliably quantify the global collective societal escalation in "how much" we are talking about climate change. Conversations happen everywhere, publicly and privately, in all manner of media, in classrooms, at dinner tables, on local news, on national news, in books, movies, pop culture, podcasts, protests, emails, press releases, in all aspects of society, in all languages. Some of the global discourse on climate change can be tracked in a semi-quantitative way using different proxies. Because of the very short nature of GC Insights papers, we have chosen to use UN climate agreements and IPCC reports as proxies for the escalation of the whole-world societal discussion of climate change, as each agreement and report is associated with an increased urgency regarding the escalating effects of climate change and the consequences of not acting. Popular media and acknowledgement of climate change and anthropogenic impacts in the public eye in turn grows is response to each agreement.

Anthro-Pokécene, lines 99-108: COP26 was in 2021 – how does the title of this summit apply to the motivations of the developers for a game released in 2013? If you are arguing for the escalation of

environmental meaning across Pokemon games, then a direct link between a Pokémon game in 2013 and a UN summit in 2021 should be provided.

*For this, we are not suggesting a link. We are highlighting that the means of combating climate change are the same for both the real and fictional worlds.*

Anthro-Pokécene, lines 123-131: It is worth noting that dystopian video games are often the opposite of nihilistic and can be hopeful & progressive – for example, see Perez-Latorre & Oliva's 2017 analysis of Bioshock Infinite. The Change citation covers the usefulness of exploring and modeling future worlds, but only covers one side without exploring how we as an audience are meant to interact with dystopic games as well.

*We have added an additional sentence to include that additional dystopian games exist that also promote hope and progressive futures.*

"*Notably, Pokémon presents a hopeful balance between humans and the environment, similar to other hopeful and progressive narrative worlds created in games (e.g. Anno 2070 ). These hopeful scenarios currently exist alongside numerous and popular nihilistic, post-apocalyptic games and stories (which can maintain underlying hopeful messages regarding humanity's ability to recover from apocalypse, despite rather bleak world views regarding the present climate crisis, e.g. Perez-Latorre & Oliva's 2017. The existence of these utopian games promotes and maintains hope that we can overcome modern environmental challenges if we want to continue to push for improvement, rather than collectively default to hopeless catastrophism.*"

---

## Editor Decision (ED1)

**General Notes**

From previous discussions, there was concern about the word count. There are several opportunities for copy editing to reduce wordiness. Example: Line 37: Opportunity for word count reduction. "… showcasing and exposing human impacts on local and global ecosystems to audiences of all ages." Example: "In order to" -> "To". While I offer some suggestions for edits and ask for additional details, I try to balance this with opportunities to balance word count. The re-wording is a suggestion, not an obligation. Please make sure that your voice is represented.

**Abstract:**

Please review and revise the abstract to ensure that the abstract matches the take home message from the article. The first sentence isn't explored so much as the perception has changed over time as a result of political discourse and media. Consider breaking the second sentence into two, as its length confounds the take home message and the journey you took to get there

**Manuscript**

Line 22-25: Recommendation to edit to something like "The public perception and societal importance of anthropogenic impacts on the environment…" and to balance word adds: Line 23 and on: "This overall perception is shaped and reflected by political discourse and news media, as well as creative and narrative media including blockbuster movies, television series, literature, and video games illustrating climate and environmental change." *Previously, it was unclear whose perception and anthropogenic impacts on or of what and there was a missing link to video games.*

Line 28: What concepts? Academic? Learning?

Line 28: To save word count from above: Recommendation to cut to something like "Research on Earth and environmental science representation in video games is still growing (Clements et al., 2022; Hut et al., 2019; McGowan & Alcott, 2022; McGowan & Scarlett, 2021), with many games set in environments inspired by real-world locations, events, and processes, making them ideal for teaching environmental concepts."

Line 34: The transition between paragraphs is a bit hard. Recommendation to bring the sentence that starts in 38 to the top of this to connect climate change to the vastness of Pokémon. Then go into Pokémon is the largest media franchise…" For example: "Since

climate change is a global challenge, Pokémon is uniquely positioned to promote climate change awareness, as it is available in 192 countries (ThePokémonCompany, 2022)."

Line 40: Recommendation to drop "In doing so,..." There isn't a clear callback to what you're referencing. Alternatively, recommendation to provide a little exposition of "To explore Pokémon's integration of climate change knowledge we ask 1) " fill in and add your questions such that the reader clearly understands why you're asking these questions, though you've done a great job setting up Pokémon's potential the "THL" should be clear here.

Line 45: Please clarify "this" since it's the start of a new section to something like "We played the main series Pokémon games released from 1996 to 2023 and thematically analyzed driving narratives and instances of anthropogenic impacts in the games to evaluate evolving anthropogenic and environmental impact themes."

Line 48: "Representative quotes were collated from each generation of 48 games by interrogating game scripts and quotes and qualitatively coding them into thematic categories, illustrated in Fig. 1. Examples can be found at link." Previously, it wasn't explicit that you had done thematic coding, the figure wasn't referenced.

Line 49: Please review the table. There are missing titles, check consistency of punctuation and capitalization, etc. I would also recommend trying to find a couple more examples just to strengthen your argument and illustrate that there isn't a "just one" example. Consider adding a column to elucidate how you find/incorporated the paragraph starting at 103 in your analysis, right now it feels detached.

Figure 1 description is incredibly long at ~100 words. Also, there are no numbered IPCC assessment reports referenced.  "Original release timeline of main-series Pokémon games and the evolution of global discourse surrounding climate change evidenced by environmental events since 1990 (e.g., climate meetings and agreements). The qualitatively coded themes of the antagonists' motives are highlighted in green."

Line 60: A bit redundant. "The modern geologic era is often referred to as the Anthropocene due to widespread human impacts across geologies and ecosystems, including climate change."

Paragraph starting at 65: Recommendation to review flow, since the good THL for this paragraph is in the sixth line of this paragraph in which the connection between the themes in the game match the decade. But, it's split up by the sentence "These games coincided...", which is a great lead in to the final paragraph that builds towards the introduction of climate change.

Line 77: "morally ambiguous" is a bit confusing here because it becomes more elucidated in the second sentence.

The paragraph starting at 77: I see that this paragraph is the transition between no climate change to more climate change; I would recommend stating that "transition phase" explicitly (however you would want to word it).

Line 80: Consider: "...to a beautifuly and unspoiled state by wiping out the population.  The sentence: "While arguably..." doesn't add additional context that greatly expands the previous line and the thought is concluded in the next sentence "This storyline mirrors...".

Line 82-83: Referring to "Without being sanctimonious..." Can you add a couple of words of context to how the player is to question the ethics? Is it just through exposure/osmosis of thought or is there a choice/action that the player must grapple with?

Line 85-87: Are there examples that pre-date the game to illustrate that the game reflects society? The current example is nearly a decade later.

Paragraph starting on Line 90: I would recommend dropping "The former region," and "latter region" because you mention them two sentences prior and you can just say the name. I read it originally as "Alola was formerly known or existed as X". It also doesn't have any cited connection to the real-life analog.

Paragraph starting on 103 is excellent. Having it in the previous section might expand the lines of thought, since the final section title and placement implies "conclusions" whereas this paragraph is giving additional discussion.

Paragraph starting on Line 115 feels like a departure from the purpose of this piece. Briefly linking it to the clash between real-life climate anxiety can link back to your driving questions. Rather than pursuing a new thesis through the exploration of nihilistic games starting in Line 119 with "These hopeful scenarios", consider using this space to link to real life discourse. That would lead well into the proceeding sentence. Same sentiment with "Games and global pheno..." rather than introducing new media, land the Pokémon thesis because the resolution of these questions should be stronger..

Line 115-116: Is this confirmed or discussed elsewhere? This is a point made in the intro, but not throughout the middle of the manuscript. Similarly, the primary sentence on Line 116-118 is a claim that isn't backed up with citations.

---

## Author Response (AR2)

General Notes

From previous discussions, there was concern about the word count. There are several opportunities for copy editing to reduce wordiness. Example: Line 37: Opportunity for word count reduction. "... showcasing and exposing human impacts on local and global ecosystems to audiences of all ages." Example: "In order to" -> "To". While I offer some suggestions for edits and ask for additional details, I try to balance this with opportunities to balance word count. The re-wording is a suggestion, not an obligation. Please make sure that your voice is represented.

*Thank you very much, we have taken your advice and attempted to reduce word count while also ensuring our voice is still represented. We have also added additional example quotes for motive in the supplementary file.*

Abstract:

Please review and revise the abstract to ensure that the abstract matches the take home message from the article. The first sentence isn't explored so much as the perception has changed over time as a result of political discourse and media. Consider breaking the second sentence into two, as its length confounds the take home message and the journey you took to get there

*We have broken the sentence into two and it now reads better. Thank you for your suggestion.*

Manuscript

Line 22-25: Recommendation to edit to something like "The public perception and societal importance of anthropogenic impacts on the environment..." and to balance word adds:

Line 23 and on: "This overall perception is shaped and reflected by political discourse and news media, as well as creative and narrative media including blockbuster movies, television series, literature, and video games illustrating climate and environmental change." Previously, it was unclear whose perception and anthropogenic impacts on or of what and there was a missing link to video games.

*Thank you very much for the suggestion. We have edited these few lines with your recommendations.*

Line 28: What concepts? Academic? Learning?

*We have clarified learning concepts*

Line 28: To save word count from above: Recommendation to cut to something like "Research on Earth and environmental science representation in video games is still growing (Clements et al., 2022; Hut et al., 2019; McGowan & Alcott, 2022; McGowan & Scarlett, 2021), with many games set in environments inspired by real-world locations, events, and processes, making them ideal for teaching environmental concepts."

*Thank you for the suggestion, we have broadly edited the sentence to your recommendation as follows: "Research into Earth and environmental science's representation in video games is still a growing field (Clements et al., 2022; Hut et al., 2019; McGowan & Alcott, 2022; McGowan & Scarlett, 2021), with many video games inspired by real world settings, events or locations, making them ideal for teaching environmental features, processes and interactions."*

Line 34: The transition between paragraphs is a bit hard. Recommendation to bring the sentence that starts in 38 to the top of this to connect climate change to the vastness of Pokémon. Then go into Pokémon is the largest media franchise…" For example: "Since climate change is a global challenge, Pokémon is uniquely positioned to promote climate change awareness, as it is available in 192 countries (ThePokémonCompany, 2022)."

We have edited the end and beginning of the two paragraphs to try and make them flow better.

Line 40: Recommendation to drop "In doing so,…" There isn't a clear callback to what you're referencing. Alternatively, recommendation to provide a little exposition of "To explore Pokémon's integration of climate change knowledge we ask 1) " fill in and add your questions such that the reader clearly understands why you're asking these questions, though you've done a great job setting up Pokémon's potential the "THL" should be clear here.

We have included the recommendation and hopefully now streamlined the sentence to better clarify the objectives of the manuscript.

Line 45: Please clarify "this" since it's the start of a new section to something like "We played the main series Pokémon games released from 1996 to 2023 and thematically analyzed driving narratives and instances of anthropogenic impacts in the games to evaluate evolving anthropogenic and environmental impact themes." Line 48: "Representative quotes were collated from each generation of 48 games by interrogating game scripts and quotes and qualitatively coding them into thematic categories, illustrated in Fig. 1. Examples can be found at link." Previously, it wasn't explicit that you had done thematic coding, the figure wasn't referenced.

We have better clarified the thematic coding and almost focused on limiting some of the unnecessary wording to limit our word count.

Line 49: Please review the table. There are missing titles, check consistency of punctuation and capitalization, etc. I would also recommend trying to find a couple more examples just to strengthen your argument and illustrate that there isn't a "just one" example. Consider adding a column to elucidate how you find/incorporated the paragraph starting at 103 in your analysis, right now it feels detached.

We have now reviewed the table and included additional examples to all of the games noted.

Figure 1 description is incredibly long at ~100 words. Also, there are no numbered IPCC assessment reports referenced. "Original release timeline of main-series Pokémon games and the evolution of global discourse surrounding climate change evidenced by environmental events since 1990 (e.g., climate meetings and agreements). The qualitatively coded themes of the antagonists' motives are highlighted in green."

We have amended the figure caption so it is now more concise. The numbered IPCC reports are also included above the timeline.

Line 60: A bit redundant. "The modern geologic era is often referred to as the Anthropocene due to widespread human impacts across geologies and ecosystems, including climate change."

We believe that a sentence stating this is needed in order when defining the Antro-Pokécene in order to ensure all readers know what we are reference from.

Paragraph starting at 65: Recommendation to review flow, since the good THL for this paragraph is in the sixth line of this paragraph in which the connection between the themes in the game match the decade. But, it's split up by the sentence "These games coincided…", which is a great lead in to the final paragraph that builds towards the introduction of climate change.

Done.

Line 77: "morally ambiguous" is a bit confusing here because it becomes more elucidated in the second sentence.

In this new revision we have edited the sentences into one to allow for a better flow.

The paragraph starting at 77: I see that this paragraph is the transition between no climate change to more climate change; I would recommend stating that "transition phase" explicitly (however you would want to word it).

We have edited the beginning of this paragraph to read as follows: *"As global climate discourse proliferated in the late 2000s and 2010s, the franchise grew and transitioned to better represent the nuance and complexity of environmental change."*

Line 80: Consider: "…to a beautifuly and unspoiled state by wiping out the population. The sentence: "While arguably…" doesn't add additional context that greatly expands the previous line and the thought is concluded in the next sentence "This storyline mirrors…".

As discussed above, we have now edited the sentences into one to allow for better flow.

Line 82-83: Referring to "Without being sanctimonious…"Can you add a couple of words of context to how the player is to question the ethics? Is it just through exposure/osmosis of thought or is there a choice/action that the player must grapple with?

We have included an additional statement to address. *"Without being sanctimonious, this concept being presented by the game's antagonist inherently causes players to question the ethics of calls to reduce human populations as a viable solution to climate change through exposure and discussion of the subject, they may not otherwise be witness to."*

Line 85-87: Are there examples that pre-date the game to illustrate that the game reflects society? The current example is nearly a decade later.

We are trying to provide direct evidence for the games and society demonstrating similar goals. We refer to the UN title and IPCC reports as they are explicitly cited as a community efforts to combat climate change.

Paragraph starting on Line 90: I would recommend dropping "The former region," and "latter region" because you mention them two sentences prior and you can just say the name. I read it originally as "Alola was formerly known or existed as X". It also doesn't have any cited connection to the real-life analog.

We have made this edit.

Paragraph starting on 103 is excellent. Having it in the previous section might expand the lines of thought, since the final section title and placement implies "conclusions" whereas this paragraph is giving additional discussion.

Thank you very much for the positive reception to this paragraph. We have moved it into the previous section as suggested.

Paragraph starting on Line 115 feels like a departure from the purpose of this piece. Briefly linking it to the clash between real-life climate anxiety can link back to your driving questions. Rather than pursuing a new thesis through the exploration of nihilistic games starting in Line 119 with "These hopeful scenarios", consider using this space to link to real life discourse. That would lead well into the proceeding sentence. Same sentiment with "Games and global pheno…" rather than introducing new media, land the Pokémon thesis because the resolution of these questions should be stronger..

We appreciate the suggestion of linking back to real life discourse. We do however believe that including the contrasting post-apocalyptic narratives to Pokémon is of value though, in an attempt to highlight that Pokémon is unique in its appreciation of the natural world.

We have now edited the text to better refer back to the questions initially posed by stating:

"Pokémon has progressed to present a more hopeful balance between humans and the environment over the past few decades. In doing so they represent how popular media has come to mirror public discourse and society aiming for a better planet, albeit whilst presenting moral dilemmas through antagonists actions."

We have also removed the statement referring to nihilistic games.

Line 115-116: Is this confirmed or discussed elsewhere? This is a point made in the intro, but not throughout the middle of the manuscript. Similarly, the primary sentence on Line 116-118 is a claim that isn't backed up with citations.

We have now provided the Rangel et al. 2022 reference referring to the scope of Pokémon in education.

---

## Author Response (AR3)

The authors have done a great job addressing many points of clarifications and editorial comments.

Thank you very much for your positive response, please see below for our responses to the two still outstanding comments.

Some of the previous major comments were not addressed, please consider addressing these in this penultimate draft.

1. Please review the figshare table. There is a missing column (though this may be a function of the table where the missing title is a second example, but that is currently not clear), check consistency of punctuation and capitalization, etc. I would also recommend trying to find a couple more examples just to strengthen your argument and illustrate that there isn't a "just one" example.

We apologise for the confusion in the table. We have updated the column heading to show there are actually at least two examples presented for anthropogenic impacts. Punctuation and capitalization of quotes is a result of the quotes themselves. We note that there are several quotes with capitals in the middle of sentences but they are just to denote shouting in the game.

2. Consider adding a column to elucidate how you find/incorporated the content in "A Hopeful World" in your analysis, right now it feels detached. There's no clear link between how your analysis in the previous section(s) link to this new direction and it starts to move away from the driving questions posed in the introduction. Alternatively, adding a sentence addressing this would meet the requirements to explain to the readers how you did your analysis and came to these conclusions. The second paragraph in this section feels like a departure from the purpose of this piece. Briefly linking it to the clash between real-life climate anxiety can link back to your driving questions, but it starts to drift. Rather than pursuing a new thesis through the exploration of nihilistic games starting with "These hopeful scenarios", consider using this space to link to real life discourse. That would lead well into the proceeding sentence. Same sentiment with "Games and global pheno…" rather than introducing new media, land the Pokémon thesis because the resolution of these questions should be stronger.
In the second paragraph, claims were made but the link to the rest of the text is missing. Is this confirmed or discussed elsewhere? This is a point made in the intro, but not throughout the middle of the manuscript and in the analysis. Similarly, the main take home lesson is a claim that isn't backed up with citations or your analysis, which would be a great way to tie these things together for the reader.

We have now provided additional "hopeful world" examples from the games including images to support the discussion in the second part of the manuscript. We have also added additional text throughout to better orientate the reader to the efforts we are making in discussing both negative and positive environmental issues and approaches.

We have now removed a large proportion of the concluding paragraph, now ended with a short summary addressing both the negative and positive representations.

---

## Author Response (AR4)

1. You have done a good job demonstrating how you conducted your qualitative analysis of the storyline themes. However, the qualitative analysis on the other items is missing. Can you share the data that you collected about the increased appearance of different features (e.g., wildlife areas, bike paths) and anthropogenic impacts? This could be in a table to complement Figure 1 to demonstrate how these have changed over time, since this is some of the data you center. This could be added to the existing Fig Share figure or created anew to be put in the text.

Thank you for the acknowledgement of our improved manuscript. While we understand the query, we do believe that we have already noted in the supplementary table, the anthropogenic impacts. We have now included additional information regarding the occurrence of wildlife areas and bike paths in an additional column.

2. Please move the Fig Share figure to supplemental so that it's static.

The link has now been moved to the Data availability section and removed from the methods.